# Evaluating the Relative Vaccine Effectiveness of Adjuvanted Trivalent Influenza Vaccine Compared to High-Dose Trivalent and Other Egg-Based Influenza Vaccines among Older Adults in the US during the 2017–2018 Influenza Season

**DOI:** 10.3390/vaccines8030446

**Published:** 2020-08-07

**Authors:** Stephen I. Pelton, Victoria Divino, Drishti Shah, Joaquin Mould-Quevedo, Mitch DeKoven, Girishanthy Krishnarajah, Maarten J. Postma

**Affiliations:** 1Department of Pediatrics, Boston University Schools of Medicine, Boston, MA 02118, USA; spelton@bu.edu; 2Maxwell Finland Laboratories, Boston Medical Center, Boston, MA 02118, USA; 3Department of Health Economics & Outcomes Research, Real-World Evidence, IQVIA, Falls Church, VA 22042, USA; drishti.shah@iqvia.com (D.S.); Mitch.DeKoven@iqvia.com (M.D.); 4Department of Global Market Access, Seqirus Vaccines Ltd., Summit, NJ 07901, USA; Joaquin.Mould-Quevedo@seqirus.com (J.M.-Q.); Shanthy.Krishnarajah@seqirus.com (G.K.); 5Unit of PharmacoTherapy, -Epidemiology & -Economics (PTE2), Department of Pharmacy, University of Groningen, 9700 Groningen, The Netherlands; m.j.postma@rug.nl; 6Department of Health Sciences, University of Groningen, University Medical Centre Groningen (UMCG), 9700 Groningen, The Netherlands; 7Department of Economics, Econometrics & Finance, University of Groningen, Faculty of Economics & Business, 9700 Groningen, The Netherlands

**Keywords:** influenza, influenza vaccine, adjuvanted influenza vaccine, relative vaccine effectiveness, elderly, retrospective studies

## Abstract

The influenza-related disease burden is highest among the elderly. We evaluated the relative vaccine effectiveness (rVE) of adjuvanted trivalent influenza vaccine (aTIV) compared to other egg-based influenza vaccines (high-dose trivalent (TIV-HD), quadrivalent (QIVe-SD), and standard-dose trivalent (TIVe-SD)) against influenza-related and cardio-respiratory events among subjects aged ≥65 years for the 2017–2018 influenza season. This retrospective cohort analysis used prescription claims, professional fee claims, and hospital charge master data. Influenza-related hospitalizations/ER visits and office visits and cardio-respiratory events were assessed post-vaccination. Inverse probability of treatment weighting (IPTW) and Poisson regression were used to evaluate the adjusted rVE of aTIV compared to other vaccines. In an economic analysis, annualized follow-up costs were compared between aTIV and TIV-HD. The study was composed of 234,313 aTIV, 1,269,855 TIV-HD, 212,287 QIVe-SD, and 106,491 TIVe-SD recipients. aTIV was more effective in reducing influenza-related office visits and other respiratory-related hospitalizations/ER visits compared to the other vaccines. For influenza-related hospitalizations/ER visits, aTIV was associated with a significantly higher rVE compared to QIVe-SD and TIVe-SD and was comparable to TIV-HD. aTIV was also associated with a significantly higher rVE compared to TIVe-SD against hospitalizations/ER visits related to pneumonia and asthma/COPD/bronchial events. aTIV and TIV-HD were associated with comparable annualized all-cause and influenza-related costs. Adjusted analyses demonstrated a significant benefit of aTIV against influenza- and respiratory-related events compared to the other egg-based vaccines.

## 1. Introduction

Influenza is a serious and contagious respiratory illness that is associated with high disease burden. The 2017–2018 flu season was a high severity season in the United States (US). According to preliminary estimates from the Centers for Disease Control and Prevention (CDC), the 2017–2018 flu season led to 45 million cases of symptomatic illness, 21 million visits to a healthcare provider, 808,000 hospitalizations, and 61,000 deaths [1].

The burden of influenza is disproportionately higher among the elderly (65 + years of age), as older adults are more vulnerable to severe influenza disease and are at a higher risk of developing influenza-related complications [2,3]. The elderly accounted for 90% of influenza-related deaths during the 2017–2018 flu season [1]. Influenza-related morbidity and mortality is highest among this age group due to immunosenescence and lower vaccine effectiveness [2,3]. Comorbidities are more common among the elderly, many of which are risk factors for influenza-related complications. Influenza can also exacerbate pre-existing, chronic comorbidities.

In the US, there are two influenza vaccines that are currently approved specifically for the ≥65 population: adjuvanted trivalent influenza vaccine (aTIV; Fluad^®^, Seqirus) and high-dose trivalent influenza vaccine (TIV-HD; Fluzone High-Dose^®^, Sanofi Pasteur) [4]. These enhanced influenza vaccines present opportunities to reduce the influenza-related burden among older adults compared to standard influenza vaccines [3,5]. aTIV combines MF59 adjuvant (an oil-in-water emulsion of squalene oil) and a standard dose of antigen, and is designed to produce stronger, broader, and longer immune responses against the selected influenza vaccine strains [3,6]. TIV-HD contains a higher concentration of antigen than the standard-dose influenza vaccines and is designed to produce stronger immune responses against the selected influenza vaccine strains [3].

Both aTIV and TIV-HD are manufactured using an egg-based process. The other egg-based influenza vaccines available during the 2017–2018 flu season in the US were a standard-dose trivalent influenza vaccine (TIVe-SD) and quadrivalent influenza vaccine (QIVe-SD). Trivalent influenza vaccines contain antigens from two influenza A viruses (H1N1 and H3N2) and one influenza B virus (B Yamagata or B Victoria) [7]. During the 2017–2018 flu season, trivalent influenza vaccines included a B/Brisbane/60/2008–like virus (Victoria lineage) [8]. Quadrivalent vaccines additionally protect against the second influenza B virus. During the 2017–2018 flu season in the US, influenza A (H3N2) viruses predominated overall; however, there was an increase in the circulation of influenza B viruses (primarily B Yamagata) starting in March 2018 [9].

In Europe, aTIV had been used for over 20 years and has demonstrated significant benefits compared to TIVe-SD in terms of reducing the relative risk of influenza-like illness and hospitalization among the elderly [6,10,11,12]. aTIV was introduced in the United Kingdom (UK) in 2018 and was recommended for use in the 65+ population during the 2018–2019 season [13]. During the 2018–2019 season in the UK, aTIV showed a benefit against laboratory-confirmed influenza, compared to subjects with any type of influenza vaccine in that age group [14].

aTIV was introduced in the US in 2015; therefore, there are limited studies evaluating the relative vaccine effectiveness (rVE) of aTIV compared to other egg-based vaccines in the US. One comprehensive analysis compared the rVE of cell-cultured quadrivalent influenza vaccine (QIVc) and egg-based influenza vaccines (aTIV, TIV-HD, QIVe-SD, and TIVe-SD), among a Medicare fee-for-service (FFS) population [15]. Another study compared the rVE of aTIV and TIV-HD using single-sourced third-party payer data [16]. Studies in the US suggest that TIV-HD is more effective than TIVe-SD at reducing clinical outcomes associated with influenza infection in older adults [17,18,19].

The objective of this analysis was to evaluate the rVE of aTIV compared to other egg-based influenza vaccines (TIV-HD, QIVe-SD, and TIVe-SD) against influenza-related hospitalizations/emergency room (ER) visits, influenza-related office visits, and cardio-respiratory hospitalizations/ER visits during the 2017–2018 flu season, among a representative elderly population, including Medicare FFS and Medicare Advantage enrollees, in the US. In an economic analysis, all-cause and influenza-related healthcare resource utilization (HCRU) and costs were compared between aTIV and TIV-HD subjects.

## 2. Materials and Methods

### 2.1. Study Overview

This retrospective cohort study was conducted among subjects ≥65 years of age who were vaccinated with aTIV, TIV-HD, QIVe-SD, or TIVe-SD during the 2017–2018 flu season in the US. A linked subject population was utilized, using de-identified data from several IQVIA databases: Professional Fee Claims (Dx), Prescription Claims (Rx) and Hospital Charge Data Master (CDM). These databases have been previously described [20,21]. Ethics approval was not required because this was a retrospective analysis of secondary data with deidentified data.

### 2.2. Data Sources

Dx includes approximately 1 billion professional fee claims per year, representing over 870,000 practitioners per month. Dx includes professional fee claims representing 60–70% of physician activity in the US. Rx includes more than 1.6 billion retail or mail-order prescription claims, representing dispensed prescriptions for approximately 85% of all pharmacies in the US. CDM includes records from hospital charge data master files, the service order records drawn from hospital operational files, and other reference sources. CDM includes records from over 450 hospitals, covering 7 million annual inpatient stays and 60 million annual outpatient visits. The datasets were linked using a deterministic matching algorithm, using actual patient information (e.g., patient age, gender, number of visits, etc.) to assign a unique patient ID. This ensured continuity of patient follow-up across datasets.

All data from IQVIA’s Dx, Rx, and CDM databases are compliant with the Health Insurance Portability and Accountability Act (HIPAA) to protect patient’s privacy. These data sources are representative of the 65 + population and include all payers (including traditional Medicare FFS).

### 2.3. Study Population

The 2017–2018 flu season was defined as beginning on 1 August 2017 (based on the observed distribution of vaccination by month) and ending on 4 August 2018 [15]. The study period began on 1 February 2017, allowing for a 6-month pre-index or baseline period, and ended on 4 August 2018.

Subjects with ≥1 medical or pharmacy claim for aTIV, TIV-HD, QIVe-SD, or TIVe-SD in Dx or Rx between 1 August 2017 and 31 January 2018 (the “selection window”) were initially identified (Figure 1). The first claim determined the vaccine cohort and the date was termed the “index date”. Cohorts were mutually exclusive. Subjects were required to be ≥65 years of age at the index date. Linkage in both Dx and Rx was required during the study period. Since the databases utilized are open-source, proxies for continuous enrollment (CE) were applied. Subject activity, defined as ≥1 office visit (in Dx) and ≥1 prescription (in Rx), was required in the 6 months prior to the 6-month pre-index period, as well as following the end of the flu season (through November 2018; the end of available data at the time of analysis). Pharmacy stability was required in the 6-month pre-index period through the end of the flu season, defined as consistent reporting of data from the pharmacy most frequently visited by the subject and ≥80% coverage rate for that pharmacy in each month.

Patients were excluded if they had an influenza-related hospitalization or ER visit or an influenza-related office visit (subsequently defined in the Study Measures section) between the beginning of the flu season up to 13 days after the index date. Subjects were excluded if they received any other flu vaccine during the 2017–2018 flu season other than the vaccine received on the index date or if they had incomplete data or data quality issues (invalid year of birth or missing gender). Finally, patients without linkage to CDM at any time were excluded.

The fixed 6-month pre-index or baseline period was used to assess the study eligibility criteria and measure subject baseline characteristics. Study outcomes were assessed over the variable post-index or follow-up period, which began 14 days after the index date (in order to allow for the development of vaccine-specific immunity) through the end of the flu season (4 August 2018).

### 2.4. Study Measures

Baseline demographic characteristics were assessed at the index date, including age, gender, geographic region, and payer type. Clinical characteristics were measured over the 6-month pre-index period (not including the index date, unless otherwise specified) and included the month of flu vaccination, Charlson comorbidity index (CCI; Dartmouth–Manitoba adaptation based on ICD-9-CM and ICD-10-CM diagnosis codes), comorbidities of interest [15], indicators of frail health status [15], and pre-index all-cause costs (outpatient pharmacy, medical (inpatient and outpatient (ER)), and total). Since Dx and CDM only include charges, a cost:charge ratio (CCR) was applied to charges, using the Centers for Medicare and Medicaid Services (CMS) hospital outpatient prospective payment system (OPPS) CCR files and the Healthcare Cost and Utilization Project (HCUP) Inpatient CCR files, respectively [22,23].

Clinical outcomes of interest were assessed starting 14 days after the index date. The number and rates (events per 1000 vaccinated-subject seasons) of the following events were assessed: influenza-related hospitalization/ER visits, influenza-related office visits, and cardio-respiratory hospitalization/ER visits. Cardio-respiratory hospitalization/ER visits related to the following events were assessed: pneumonia, asthma/COPD/bronchial, coronary artery (including myocardial infarction), congestive heart failure, cerebrovascular (including stroke), and other respiratory events (e.g., acute and chronic sinusitis, laryngitis, lower and upper respiratory tract infection, etc.). Serious adverse events (SAEs) related to these cardio-respiratory events have previously been evaluated in a multicenter trial of patients vaccinated with either TIV-HD or standard-dose egg-based vaccines [24]. These cardio-respiratory events were included in this analysis because secondary complications, such as pneumonia or bronchitis, and exacerbations of chronic medical conditions, cause considerable influenza-related morbidity and mortality [2]. A vaccine that is effective at preventing influenza may in turn reduce influenza-related complications or influenza-related exacerbations of pre-existing conditions. Furthermore, cardiovascular disease (CVD) is the most commonly identified chronic medical condition among adults hospitalized with influenza, and influenza vaccination has been shown to reduce the incidence of major adverse cardiac events (MACE) among those with existing CVD [25]. For each outcome of interest, the first occurring event was identified. A subject could contribute an event for more than one outcome but could not contribute more than one event to the same outcome.

Influenza-related hospitalizations and ER visits were defined as a hospitalization or ER visit with a diagnosis code for influenza (ICD-9 487.x, 488.x, ICD-10 J09.x, J10.x, J11.x) in any position. Influenza-related office visits were defined based on a physician office visit with a claim for a rapid influenza diagnostic test (CPT code 87804) and with a prescription fill for a treatment regimen of oseltamivir (75 mg twice daily for 5 days; capsule formulation only) within 2 days of the test. Definitions for these influenza-related events followed similar published methods [15]. Cardio-respiratory hospitalization/ER visits were defined based on a hospitalization or ER visit with a diagnosis code in any position for the cardio-respiratory event of interest.

### 2.5. Analyses

Descriptive statistics were reported for each study cohort. Standardized mean differences (SMD) were calculated to evaluate the difference in baseline covariates between the vaccine cohorts relative to aTIV. SMD was calculated as the difference in means or proportions of a variable divided by the pooled standard deviation. SMD (absolute) of ≥0.10 between groups was considered statistically meaningful [26].

Our statistical approach followed similar published methods [15]. Since unadjusted comparisons between the vaccine cohorts may be confounded by treatment selection bias, inverse probability of treatment weighting (IPTW) was used to adjust for confounders and treatment selection bias. An individual pairwise comparison approach was taken versus aTIV. A pseudo-population was created, composed of individuals in the pre-IPTW population weighted by the inverse of their probability of receiving aTIV, given the baseline covariates. Weights were constructed by estimating each subject’s probability of receiving aTIV based on observed covariates in a logistic regression model. Separate models were developed for each pairwise comparison versus aTIV. Clinically relevant baseline variables with SMD ≥0.10 in the unadjusted sample were included in the models as independent variables. The propensity score for each individual (the predicted probability of receiving aTIV) was estimated. Unstabilized weights were calculated as the inverse of the propensity score for a subject. A stabilized IPTW approach was utilized in order to reduce type 1 error [27,28]. To calculate stabilized weights, the numerator of the unstabilized weights was replaced by the marginal probability of receiving aTIV over the other vaccine. Additionally, weight values greater than five were truncated to five due to the potential bias of outliers. Baseline characteristics and SMD post-IPTW were reported as a measure of balance. The clinical outcomes of interest were evaluated for the IPTW sample.

Poisson regression models were used to permit a more robust regression adjustment as well as to further reduce bias due to residual confounding. IPTW-weighted multivariate Poisson regression models were developed to estimate adjusted rate ratios (RR) along with corresponding 95% confidence intervals (CIs) for aTIV compared to the other vaccines in pairwise comparisons. Adjusted rVE was calculated as (1-RR) × 100% along with the corresponding 95% CIs. The models included variables that were considered clinically relevant but that were not included in the IPTW because they were well balanced (SMD <0.10) in the unmatched sample.

All analyses were based on observed, not projected, data. As all planned comparisons and their corresponding *p*-values are presented, and this was not a confirmatory study, no mathematical correction was employed for multiple comparisons [29,30,31]. Analyses were conducted using SAS^®^ Release 9.4 (SAS Institute Inc., Cary, NC, USA).

### 2.6. Economic Analysis

An economic analysis was conducted among aTIV and TIV-HD only. This comparison was considered relevant as these influenza vaccines are the only influenza vaccines with specific indications for subjects ≥65 years of age and that are primarily used in the elderly population in the US. Propensity score matching (PSM) was used to adjust for measured confounders, creating more comparable groups. PSM is a common regression modeling technique used in analyses of observational data to adjust for differences between study cohorts, particularly for economic evaluations [32]. The propensity score for each individual was estimated using a logistic regression model as the probability of receiving aTIV. A greedy nearest neighbor matching technique without replacement at a ratio of 1:1 was performed, using caliper widths of 0.1 of the standard deviation of the logit of the propensity score. Baseline characteristics with SMD ≥0.10 were included in the match.

All-cause and influenza-related HCRU and costs were evaluated over the variable follow-up period, starting 14 days after the index date through the end of the flu season and therefore did not include the vaccine cost. Influenza-related HCRU and associated costs were assessed and specific to the previously defined influenza-related hospitalizations/ER visits and influenza-related office visits. Note that unlike the clinical analysis, all occurring events of the same type contributed to the total cost for a subject (e.g., first and subsequent influenza-related hospitalizations). Utilization and costs were calculated on a per patient basis, averaged across the cohort. Pairwise comparisons were made between HCRU/cost outcomes using a paired *t*-test (mean) and the Wilcoxon signed-rank test for continuous variables (median) and McNemar’s test for categorical variables. Generalized estimating equation models (GEEs) were developed among the post-PSM sample to estimate predicted costs using a recycled predictions approach [33]. The GEEs allowed for a more robust regression adjustment as well as to further reduce bias due to residual confounding.

Predicted annualized mean costs were generated for the following outcomes of interest: (1) all-cause total healthcare costs, (2) influenza-related hospitalization costs, (3) influenza-related ER costs, (4) influenza-related office visit + oseltamivir costs, and (5) influenza-related total costs. For the first outcome, a GEE with log link function and gamma distribution was developed, and outliers were adjusted for by capping post-index annualized cost at the 99th percentile [34]. Due to the rarity of an influenza-related event, two-part GEE models were developed for the remaining outcomes. The first GEE had a binomial distribution and logit link to estimate odds of having a non-zero cost for the outcome of interest (i.e., of having the outcome). The second GEE had a gamma distribution and log link to estimate the cost of the outcome of interest, among patients with the outcome of interest. Adjustment for outliers was made by capping cost at the 99th percentile among patients with at least 1 such outcome. Predicted recycled means were obtained from the parameter estimates of GEEs, and 95% CIs were obtained through bootstrapping (500 replications). Independent variables in the models included variables that were considered clinically relevant but that were not included in the PSM because they were well balanced (SMD <0.10) in the unmatched sample. Multicollinearity was evaluated during model development.

## 3. Results

### 3.1. Study Sample

The starting sample comprised a total of 37,168,480 recipients of a flu vaccine of interest (1,910,263 aTIV, 9,423,591 TIV-HD, 19,808,442 QIVe-SD, and 6,026,184 TIVe-SD; Figure 1). The final sample comprised only subjects 65 years of age or older; 234,313 aTIV subjects (12.3% of the starting cohort), 1,269,855 TIV-HD subjects (13.5%), 212,287 QIVe-SD subjects (1.1%), and 106,491 TIVe-SD subjects (1.8%). Most aTIV (766,290) and TIV-HD (3,379,197) subjects were excluded due to the patient activity and pharmacy stability requirements (40.1% and 35.9%, respectively) while the vast majority of QIVe-SD (18,325,126) and TIVe-SD (9,535,371) subjects were excluded due to being younger than 65 years of age at index (92.5% and 87.0%, respectively). Subjects had a median of 10 months of follow-up.

### 3.2. Patient Characteristics

See Appendix A for unadjusted subject baseline demographic and clinical characteristics. Several baseline characteristics were imbalanced with (absolute) SMD ≥0.1 prior to IPTW. aTIV subjects were older than QIVe-SD subjects. For example, fewer aTIV subjects were 65–74 years of age compared to QIVe-SD subjects (50.2% vs. 56.7%). There was variation in geographic region and payer type. For example, more aTIV subjects were located in the south (53.4%) compared to the other cohorts (39.7–43.3%), and fewer aTIV subjects had third-party insurance (21.3%) compared to the other cohorts (34.1–48.6%). There was also variation in month of vaccination. aTIV subjects also had a lower comorbidity burden (e.g., CCI score) compared to QIVe-SD and TIVe-SD subjects. Post-IPTW, aTIV subjects were well balanced to TIV-HD, QIVe-SD, and TIVe-SD subjects, respectively. See Table 1 and Table 2 for subject baseline demographic and clinical characteristics post-IPTW.

### 3.3. Clinical Outcomes

#### 3.3.1. Influenza-Related Hospitalizations/ER Visits and Office Visits

Event rates post-IPTW can be found in Figure 2, Figure 3 and Figure 4 and rVE can be found in Figure 5, Figure 6 and Figure 7. Following IPTW and Poisson regression adjustment, aTIV was associated with a significantly higher rVE, compared to the other vaccine cohorts, against influenza-related office visits (rVE, aTIV vs. TIV-HD 16.6% (95% CI: 10.8–22.0%); aTIV vs. QIVe-SD 36.3% (95% CI: 31.0–41.2%); and aTIV vs. TIVe-SD 25.0% (95% CI: 17.0–32.2%)).

aTIV was also significantly more effective than QIVe-SD and TIVe-SD in preventing influenza-related hospitalizations/ER visits (rVE, aTIV vs. QIVe-SD 8.6% (95% CI: 1.2–15.6%) and aTIV vs. TIVe-SD 11.2% (95% CI: 2.3–19.4%)). aTIV and TIV-HD were comparable in preventing influenza-related hospitalizations/ER visits.

#### 3.3.2. Cardio-Respiratory Hospitalizations/ER Visits

aTIV was associated with a significantly higher rVE, compared to the other vaccine cohorts, against other respiratory hospitalizations/ER visits (rVE, aTIV vs. TIV-HD 2.4% (95% CI: 0.7–4.0%); aTIV vs. QIVe-SD 4.0% (95% CI: 1.9–6.2%); and aTIV vs. TIVe-SD 7.2% (95% CI: 4.6–9.7%)). In addition, aTIV was significantly more effective than TIVe-SD in preventing pneumonia hospitalizations/ER visits (rVE, 7.6% (95% CI: 3.7–11.3%)), as well as asthma/COPD/bronchial hospitalizations/ER visits. All other comparisons versus aTIV were comparable (see Figure 5, Figure 6 and Figure 7).

### 3.4. Economic Outcomes

For the economic analysis, 234,313 aTIV subjects were matched to 234,313 TIV-HD subjects using PSM. Subjects were well balanced on all baseline characteristics following PSM. TIV-HD subjects had a significantly higher proportion with ≥1 influenza-related office visit over the variable follow-up compared to aTIV (0.54% vs. 0.43%, *p* < 0.0001). Annualized all-cause and influenza-related costs following PSM and GEE adjustment were generally consistent with observed costs post-PSM. Following GEE adjustment, aTIV and TIV-HD were associated with comparable predicted mean annualized costs for all-cause total healthcare costs, influenza-related hospitalizations costs, influenza-related ER visits costs, and influenza-related total healthcare costs (Table 3). TIV-HD was associated with significantly higher mean total annualized influenza-related office visit costs (including oseltamivir costs) among all subjects compared to aTIV subjects ($1.36 (95% CI: $1.29–$1.45) vs. $1.10 (95% CI: $1.04–$1.17)).

## 4. Discussion

This analysis utilized claims and hospital data to compare the rVE of aTIV to other egg-based vaccines (TIV-HD, QIVe-SD, and TIVe-SD) among a representative elderly population during the 2017–2018 flu season in the US. Following the pairwise IPTW adjustment, aTIV was well balanced to each of the other vaccine cohorts across the measured baseline characteristics. Following IPTW-weighted, multivariate Poisson regression, aTIV was associated with a significantly higher rVE against influenza-related office visits, and other respiratory events compared to the other egg-based vaccines (TIV-HD, QIVe-SD, and TIVe-SD). In addition, aTIV was significantly more effective than QIVe-SD and TIVe-SD in preventing influenza-related hospitalizations/ER visits. aTIV also demonstrated a significant benefit compared to TIVe-SD for pneumonia hospitalization/ER visits and asthma/COPD/bronchial hospitalization/ER visits. All-cause and influenza-related costs were comparable between aTIV and TIV-HD, with the exception of costs for influenza-related office visits, which were significantly higher for TIV-HD subjects.

For the 2017–2018 flu season in the US, the CDC has estimated a 40% overall vaccine effectiveness of the flu vaccine; however, estimates are not available by vaccine type [9]. It was not within the scope of the current study to evaluate absolute vaccine effectiveness, as our focus was on rVE. It is important to consider our study findings in the context of the virological characteristics of the 2017–2018 flu season. There was a lineage-level mismatch between trivalent influenza vaccines (B/Victoria) and circulating strains of influenza B viruses (B/Yamagata) during the 2017–2018 flu season [8,9]. However, 2017–2018 was an A(H3N2)-predominant season and influenza B viruses (primarily B/Yamagata) only predominated from March onward [8]. Mismatch from March onward could potentially result in a lower vaccine effectiveness for the trivalent vaccines (aTIV, TIV-HD, and TIV-SD), but this mismatch would presumably have an equal impact and would not impact the comparative rVE. It is plausible that vaccine effectiveness could be higher for QIVe-SD relative to aTIV due to this mismatch from March onward. However, we found that aTIV was associated with a significantly higher rVE against influenza-related outcomes compared to QIVe-SD suggesting an overall minimal impact from the mismatch. Another factor that may impact vaccine effectiveness of these egg-based vaccines is the adaptation of influenza virus to growth in eggs [35]. Note that the majority of the influenza viruses collected from the US during the 2017–2018 flu season were antigenically and genetically similar to the cell-grown reference viruses representing the 2017–2018 Northern Hemisphere influenza vaccine viruses, suggesting again a minimal impact of egg-adaptation [9].

The authors have only identified two retrospective cohort studies, which have compared the rVE of aTIV to other influenza vaccines in the US [15,16]. One recent study, Izurieta et al. used a Medicare FFS claims dataset to evaluate the rVE of mammalian cell-based vs. egg-based influenza vaccines during the 2017–2018 flu season [15]. Following IPTW and Poisson regression, aTIV was associated with a significantly higher rVE compared to QIVe-SD and TIVe-SD for influenza-related hospitalizations/ER visits (rVE, aTIV vs. QIVe-SD 3.9% (95% CI: 1.4–6.3%); aTIV vs. TIVe-SD 3.6% (95% CI: 0.7–6.4%)) [15]. While these results are similar to the current study, the remaining results differ for the rVE of aTIV for influenza-related hospitalizations/ER visits vs. TIV-HD, and the rVE for influenza-related office visits for aTIV against the other three vaccines. Izurieta reported that TIV-HD was associated with a significantly higher rVE than aTIV against influenza-related hospitalizations/ER visits (5.3% (95% CI: 3.3–7.3%)) [15]; this analysis found that the rVE against influenza-related hospitalizations/ER visits was comparable between TIV-HD and aTIV. In addition, Izurieta found aTIV had significantly lower rVE against influenza-related office visits compared to the other egg-based vaccines (rVE, aTIV vs. TIV-HD −6.83% (95% CI: −8.9–−4.6%); aTIV vs. QIVe-SD −6.6% (95% CI: −9.7–−3.5%); and aTIV vs. TIVe-SD −11.9% (95% CI: −15.9–−8.1%)) [15]. We found the reverse. The reasons for these differing findings are unknown. While we followed similar definitions and methods, our sample was not restricted to Medicare FFS patients, and post-IPTW, approximately one-third of our cohorts had a third-party payer. It is well understood that patients vary by insurance status in terms of sociodemographic and clinical characteristics such as race, age, income status, presence of comorbid conditions, receipt of treatment, and health outcomes [36,37,38]. For example, one study showed that patients with Medicare insurance constitute a greater percentage of influenza-related inpatient stays as compared to influenza-related ED visits [36]. Studies have found that the prevalence of obesity, diabetes, and other chronic conditions is higher in Medicare beneficiaries as compared to commercially insured adults [37]. Therefore, it is plausible that the inherent variations in sample characteristics between the two studies could be associated with differences in study outcomes. While we can identify patients with Medicare or Commercial payer type in the utilized data sources, we are unable to distinguish those with Medicare Advantage. Therefore, there may be overlap between these two groups, preventing a thorough subgroup analysis by payer type in our study.

Another recent study found an rVE of 12% (95% CI: 2.1–21%) for TIV-HD vs. aTIV in reducing respiratory hospitalizations and an rVE of 6% (95% CI: 0.6–11%) in reducing cardio-respiratory hospitalizations during the 2017–2018 flu season in the US [16]. In our study, we found that while rVE against hospitalizations/ER visits related to pneumonia, asthma/COPD/bronchial, coronary artery, myocardial infarction, congestive heart failure, cerebrovascular, and stroke events were comparable between aTIV and TIV-HD, aTIV was associated with a significantly higher rVE against hospitalizations/ER visits related to other respiratory events. There are notable differences between the study of Van Aalst and colleagues limiting comparisons. Differences include data source, study population, study design, and methodology. Van Aalst et al. used Optum’s Clinformatics Data Mart Database [16], which only includes privately insured individuals and is single-payer sourced. The database does not include any traditional Medicare patients and would not be considered representative of the 65+ population. The primary outcome of the study of Van Aalst et al. was hospitalizations with a primary discharge diagnosis of ICD-10 J.x, which broadly covers all diseases of the respiratory system, while a secondary outcome was any hospitalization for a cardio-respiratory condition (ICD-10 I.x-J.x; which broadly covers diseases of the circulatory and respiratory systems). They did not look at cardiovascular outcomes separately. In addition, Van Aalst et al. utilized the previous event rate ratio (PERR) method to adjust for unobserved baseline confounders [16], but this methodological approach has its own limitations. For instance, PERR adjustment will be biased if prior events modify the risk of subsequent exposure [39]. For the primary outcome, Van Aalst et al. adjusted for the relative risk of respiratory-related hospitalizations during a baseline period from July 1 to date of flu vaccination [16]. PERR requires a number of assumptions about constant temporal effects, yet the influenza-related burden will be highest late fall/winter. Using a broader period can produce bias towards the null arising from misclassification of study outcomes [18]. The Joint Committee on Vaccination and Immunisation (JCVI) in the UK recently concluded that it could not ascertain an advantage between aTIV and TIV-HD based on the available evidence [40]. Other recent studies conducted in the US have shown a higher rVE for TIV-HD compared to TIVe-SD [18,19].

The findings from the current study are consistent with prior studies from Europe, where aTIV has been approved for more than two decades [6]. For example, a retrospective nested case-control analysis using an Italian database of electronic patient records across 15 consecutive influenza seasons (2002–2016) found that aTIV was associated with a 39% reduction in the risk of hospitalizations due to the composite of pneumonia, stroke, or myocardial infarction compared to TIVe-SD [12]. Another Italian study found that aTIV was associated with a 25% reduction in the risk of hospitalization due to pneumonia or influenza across the 2006–2009 flu seasons compared to TIVe-SD [11]. Similarly, we found aTIV to be significantly more effective in preventing hospitalizations/ER visits related to pneumonia compared to TIVe-SD; however, stroke and myocardial infarction outcomes were comparable between the two vaccines. For the elderly in the UK, Public Health England estimated that aTIV was associated with an adjusted vaccine effectiveness of 62.0% against laboratory-confirmed influenza in 2018–2019, compared to an adjusted vaccine effectiveness of 49.9% among subjects with any type of influenza vaccine [14]. Note that comparative European data is not available for aTIV vs. TIV-HD in Europe, as TIV-HD was only approved in the UK in 2019 but has not been approved in the rest of Europe [41].

As only aTIV and TIV-HD are exclusively indicated in the US for older adults, the current study also compared their economic outcomes in an economic analysis. Following PSM and GEE adjustment, we found that vaccination with aTIV or TIV-HD was associated with comparable all-cause and influenza-related annualized costs. While aTIV was associated with significantly lower annualized influenza-related office visit costs, these events were infrequent, and had relatively lower costs compared to influenza-related hospitalizations or ER visits. More real-world studies are needed to further explore potential cost-savings associated with these vaccines using different data sources, study populations, and different flu seasons.

Our study is subject to the typical limitations associated with retrospective database studies [42]. There are several limitations of this study specific to the study design and data sources utilized that are worth noting. Our study estimates are obtained from a sample of adults using linked claims and hospital data. It is an important limitation of the current study that the presence of influenza could not be confirmed using laboratory test results due to the absence of this clinical detail in the utilized databases. Future research should consider performing sensitivity analyses among the subgroups of patients with performed lab tests and with available test results (where claims and hospital data can be linked to laboratory data). A specific and established definition was used to identify influenza-related office visits (also termed probable influenza infection) based on an office visit with a claim for a rapid influenza diagnostic test and a prescription fill for a treatment regimen of oseltamivir [15,43]. It is possible that influenza-related office visit outcomes would be different if the definition was based on a diagnosis code, similar to the definitions used for hospitalizations/ER visits related to influenza and cardio-respiratory events. Additionally, the influenza-related office visit definition may be sensitive to differences in patient health seeking behaviors as well as physician treatment preferences.

There are also limitations related to the utilization of open-source databases. It is plausible that the study findings may be affected by unobservable and unmeasured factors that are not captured through the claims data. Nevertheless, patients were well balanced on all measured confounders available in the data. There is a loss of visibility to healthcare activity/consumption outside of participating pharmacies/offices/hospitals in the databases. Not all healthcare resource utilization or costs may be captured, limiting a comprehensive analysis of clinical or economic outcomes. Linkage to CDM may bias the study sample towards a more severe population, due to the requirement of having a hospitalization in CDM at any time during the available CDM data. However, patients were required to have activity following the end of the flu season, meaning that any patients with mortality during the flu season were excluded. Therefore, we provide a conservative estimate of the clinical outcomes of interest since we are unable to include influenza-related events, which lead to mortality. Despite these limitations, our study has important strengths. We used robust methodology to adjust for confounders and estimate adjusted rVE. Finally, our sample is representative of the 65+ population in the US and was not restricted to a single payer; a key limitation of other real-world studies.

## 5. Conclusions

In adjusted analyses, we found that aTIV was significantly more effective in preventing influenza-related office visits and other respiratory hospitalization/ER events compared to TIV-HD, QIVe-SD and TIVe-SD. aTIV was also associated with higher rVE against influenza-related hospitalization/ER visits compared to QIVe-SD and TIVe-SD, and was similar compared to TIV-HD. aTIV was also associated with higher rVE against hospitalizations/ER visits related to pneumonia and asthma/COPD/bronchial events compared to TIVe-SD. All-cause and influenza-related total costs were comparable between aTIV and TIV-HD.

## Figures and Tables

**Figure 1 vaccines-08-00446-f001:**
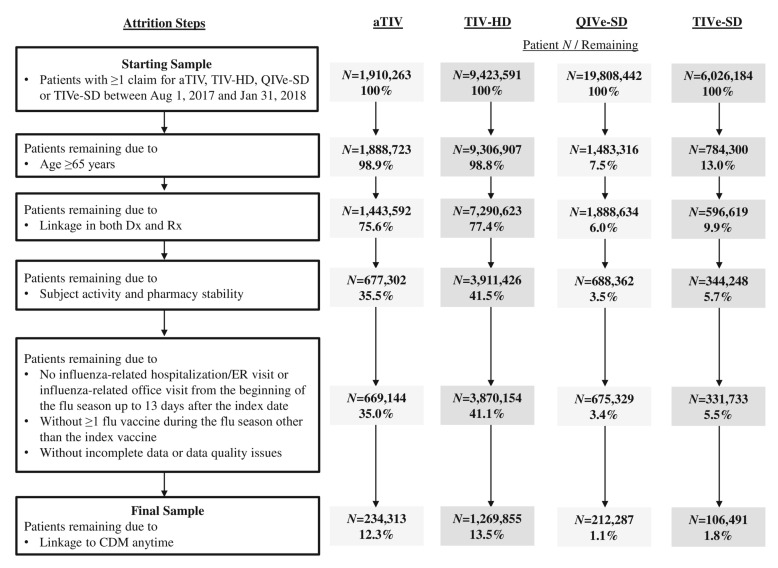
Patient selection. aTIV: adjuvanted trivalent influenza vaccine; CDM: Hospital Charge Data Master; Dx: Professional fee claims; QIVe-SD: quadrivalent influenza vaccine; Rx: Prescription claims: TIV-HD: high-dose trivalent influenza vaccine; TIVe-SD: standard-dose trivalent influenza vaccine.

**Figure 2 vaccines-08-00446-f002:**
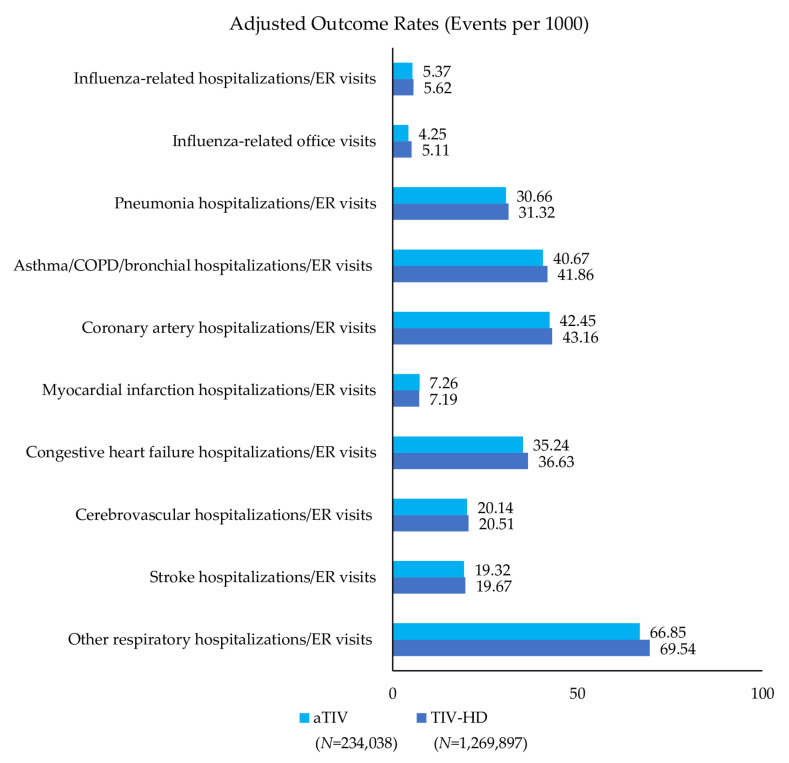
Adjusted outcome rates—post-IPTW—adjuvanted trivalent influenza vaccine (aTIV) vs. high-dose trivalent influenza vaccine (TIV-HD). Rate = events per 1000 vaccinated-subject seasons.

**Figure 3 vaccines-08-00446-f003:**
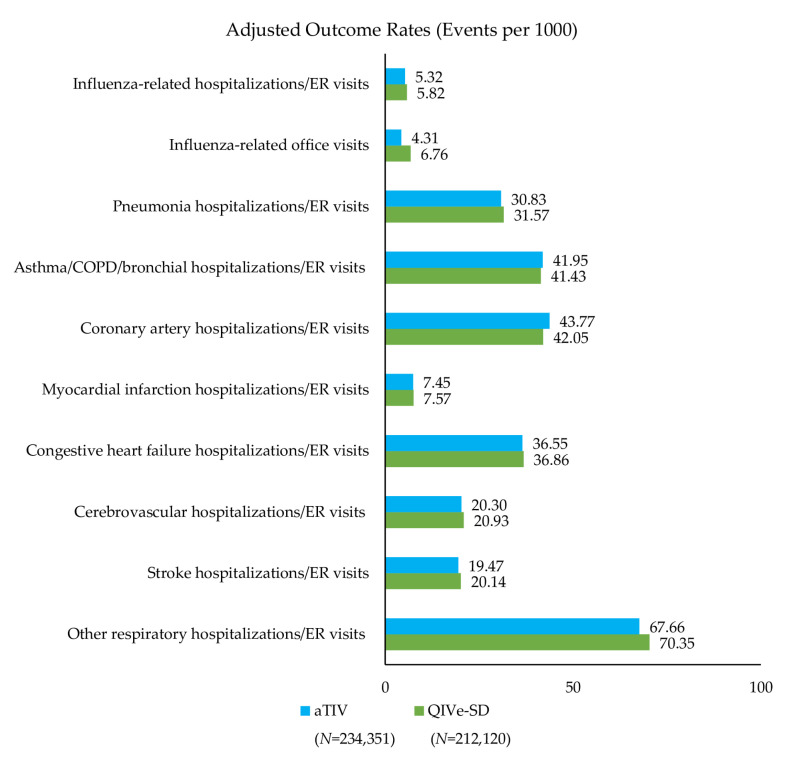
Adjusted outcome rates—post-IPTW—aTIV vs. quadrivalent influenza vaccine (QIVe-SD). Rate = events per 1000 vaccinated-subject seasons.

**Figure 4 vaccines-08-00446-f004:**
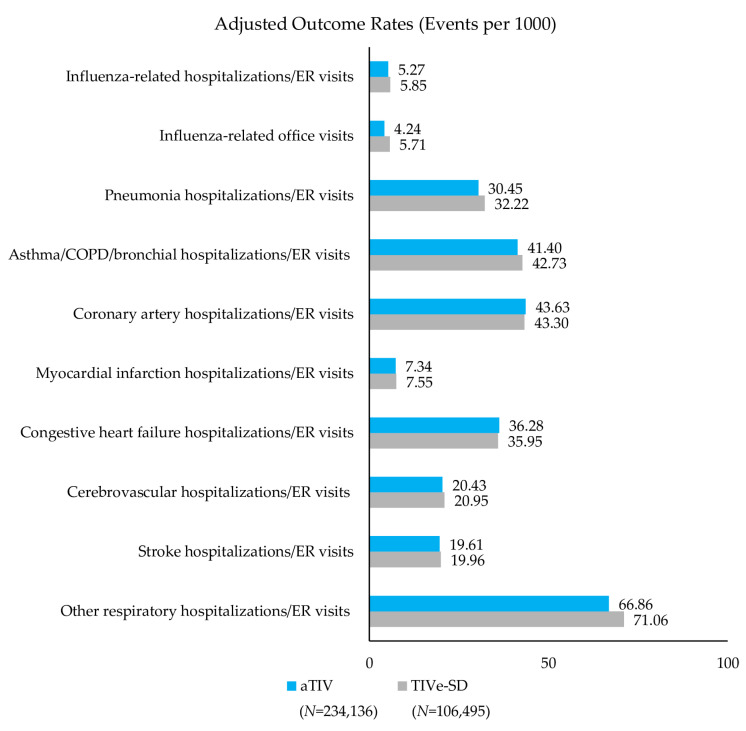
Adjusted outcome rates—post-IPTW—aTIV vs. standard-dose trivalent influenza vaccine (TIVe-SD). Rate = events per 1000 vaccinated-subject seasons.

**Figure 5 vaccines-08-00446-f005:**
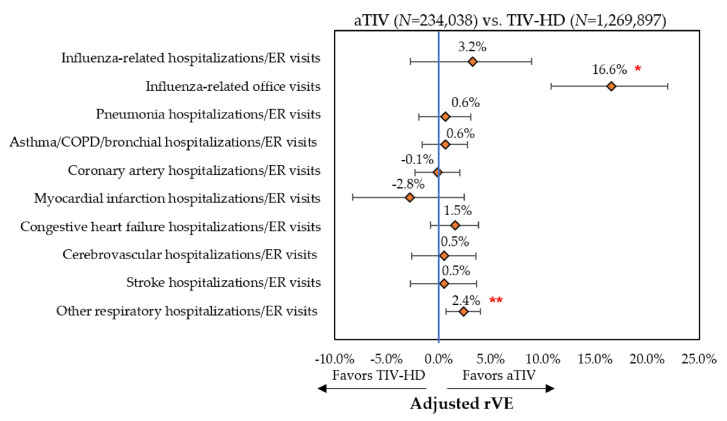
Adjusted relative vaccine effectiveness (rVE)—post-IPTW and Poisson regression—aTIV vs. TIV-HD. * *p* < 0.001; ** *p* < 0.01; rVE = relative vaccine effectiveness.

**Figure 6 vaccines-08-00446-f006:**
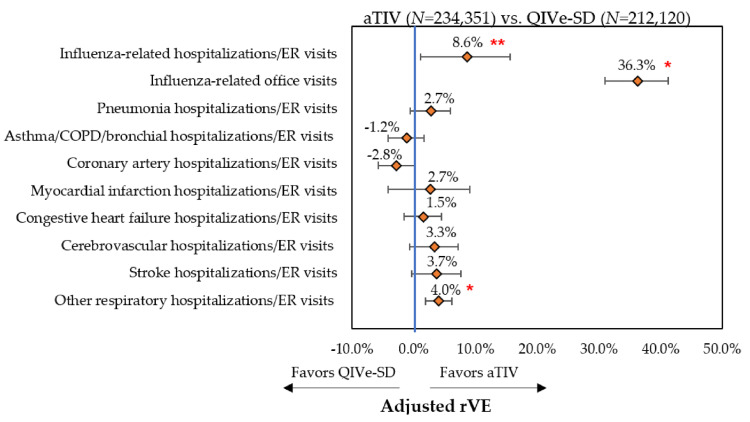
Adjusted rVE—post-IPTW and Poisson regression—aTIV vs. QIVe-SD. * *p* < 0.001; ** *p* < 0.05; rVE = relative vaccine effectiveness.

**Figure 7 vaccines-08-00446-f007:**
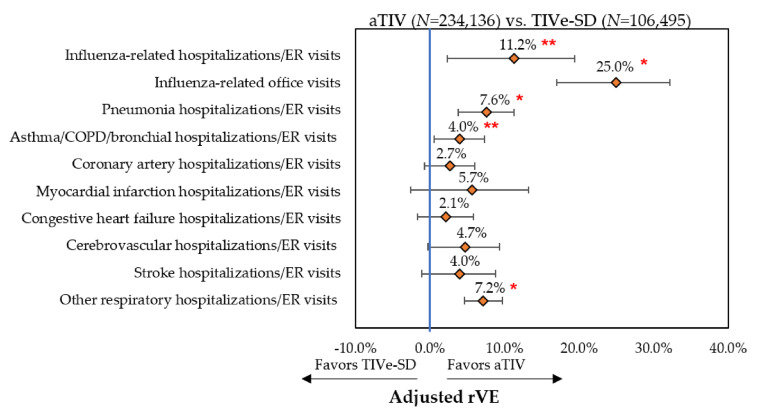
Adjusted rVE—post-IPTW and Poisson regression—aTIV vs. TIVe-SD. * *p* < 0.001; ** *p* < 0.05; rVE = relative vaccine effectiveness.

**Table 1 vaccines-08-00446-t001:** Baseline demographic characteristics—post-inverse probability of treatment weighting (IPTW).

Vaccine Cohort	aTIV	TIV-HD	SMD ^1^	aTIV	QIVe-SD	SMD ^1^	aTIV	TIVe-SD	SMD ^1^
Characteristic	*N* = 234,038	*N* = 1,269,897	-	*N* = 234,351	*N* = 212,120	-	*N* = 234,136	*N* = 106,495	-
Mean Age	75.0	75.0	0.01	74.8	74.4	−0.06	75.1	74.6	−0.09
SD	6.3	6.3	-	6.3	6.7	-	6.3	6.5	-
Median	74	74	-	74	74	-	74	74	-
Age Group (%)									
65–74 years	51.0%	50.7%	−0.01	53.2%	53.2%	0.00	50.3%	53.3%	0.06
75–84 years	33.4%	34.0%	0.01	31.7%	31.7%	0.00	33.8%	31.6%	−0.05
≥85 years	15.6%	15.3%	−0.01	15.1%	15.1%	0.00	15.9%	15.1%	−0.02
Gender (%)									
Female	59.6%	59.9%	0.01	59.3%	60.6%	0.03	59.5%	61.7%	0.04
Geographic Region (%)									
Northeast	17.3%	16.9%	−0.01	15.9%	15.9%	0.00	14.8%	15.1%	0.01
Midwest	17.4%	17.7%	0.01	14.7%	14.7%	0.00	11.3%	11.8%	0.02
South	44.9%	44.8%	0.00	47.1%	46.8%	−0.01	50.3%	49.7%	−0.01
West	20.4%	20.5%	0.00	22.3%	22.6%	0.01	23.6%	23.4%	−0.01
Payer Type (%)									
Third party	32.2%	32.1%	0.00	34.2%	34.3%	0.00	28.5%	28.6%	0.00
Medicare Part D	29.6%	29.5%	0.00	24.9%	25.0%	0.00	28.0%	28.8%	0.02
Medicare	37.6%	37.7%	0.00	39.7%	39.6%	0.00	42.4%	41.5%	−0.02
Other/Unknown	0.7%	0.7%	0.00	1.2%	1.1%	0.00	1.2%	1.1%	−0.01

^1^ SMD (absolute) ≥0.1 indicates significance; SMD = standardized mean difference.

**Table 2 vaccines-08-00446-t002:** Baseline clinical characteristics—post-IPTW.

Vaccine Cohort	aTIV	TIV-HD	SMD ^1^	aTIV	QIVe-SD	SMD ^1^	aTIV	TIVe-SD	SMD ^1^
Characteristic	*N* = 234,038	*N* = 1,269,897		*N* = 234,351	*N* = 212,120		*N* = 234,136	*N* = 106,495	
Month of Flu Vaccination (%)									
August	6.0%	6.1%	0.00	6.1%	6.0%	−0.01	6.3%	5.7%	−0.03
September	30.4%	30.9%	0.01	26.6%	26.6%	0.00	25.5%	25.2%	−0.01
October	41.1%	41.1%	0.00	40.9%	41.0%	0.00	41.2%	41.6%	0.01
November	14.3%	14.0%	−0.01	15.4%	15.3%	0.00	15.5%	15.6%	0.00
December	4.7%	4.5%	−0.01	5.9%	5.9%	0.00	5.9%	6.0%	0.01
January	3.5%	3.4%	−0.01	5.1%	5.2%	0.00	5.7%	5.8%	0.01
CCI Score (%)									
0	56.9%	55.1%	−0.04	53.3%	53.5%	0.00	53.6%	53.8%	0.00
1	20.8%	21.4%	0.01	21.6%	21.5%	0.00	21.4%	21.3%	0.00
2	11.9%	12.3%	0.01	12.8%	12.8%	0.00	12.8%	12.7%	0.00
3+	10.4%	11.3%	0.03	12.4%	12.2%	0.00	12.2%	12.2%	0.00
Mean CCI Score	0.9	0.9	0.04	1.0	1.0	0.00	0.9	1.0	0.01
SD	1.3	1.3	-	1.4	1.4	-	1.4	1.4	-
Median	0	0	-	0	0	-	0	0	-
Pre-index Comorbidities (%)									
Asthma	3.6%	3.8%	0.01	4.0%	3.9%	−0.01	3.9%	4.0%	0.00
Blood Disorders	0.3%	0.3%	0.00	0.3%	0.3%	0.00	0.3%	0.3%	0.00
Chronic Lung Disease	8.4%	8.8%	0.01	9.1%	9.5%	0.01	9.0%	9.8%	0.03
Diabetes	20.0%	21.6%	0.04	22.1%	23.6%	0.03	21.8%	24.1%	0.06
Heart Disease	12.1%	12.5%	0.01	13.3%	13.2%	0.00	13.3%	13.0%	−0.01
Kidney Disorders	8.2%	8.8%	0.02	9.5%	8.8%	−0.02	9.5%	9.0%	−0.02
Liver Disorders	2.1%	2.1%	0.00	2.4%	2.4%	0.00	2.3%	2.4%	0.00
Neurological or Neurodevelopmental Conditions	4.8%	4.8%	0.00	5.0%	5.1%	0.01	5.0%	4.8%	−0.01
Weakened Immune system ^2^	9.7%	9.9%	0.01	9.8%	9.8%	0.00	9.8%	9.4%	−0.01
IBD	0.6%	0.6%	0.00	0.6%	0.6%	0.00	0.6%	0.6%	0.00
Indicators of Frail Health Status (%)									
Home oxygen use	4.1%	4.0%	0.00	4.3%	4.4%	0.00	4.3%	4.4%	0.00
Wheelchair use	2.2%	2.3%	0.01	2.3%	2.7%	0.03	2.3%	2.5%	0.02
Walker use	3.3%	3.3%	0.00	3.4%	3.5%	0.01	3.4%	3.5%	0.01
Dementia	1.3%	1.3%	0.00	1.4%	1.2%	−0.01	1.4%	1.3%	0.00
Urinary catheter use	0.4%	0.4%	0.00	0.5%	0.4%	−0.01	0.5%	0.4%	−0.02
Falls	0.9%	0.9%	0.00	0.9%	0.8%	−0.01	0.9%	0.9%	0.00
Fractures	0.6%	0.6%	0.00	0.6%	0.5%	−0.01	0.6%	0.6%	0.00
Pre-index hospitalization (%)	7.8%	7.9%	0.00	8.2%	7.8%	−0.01	8.2%	7.7%	−0.02
Mean pre-index outpatient pharmacy costs	$2398	$2430	0.01	$2473	$2323	−0.03	$2449	$2479	0.01
SD	$5924	$5488		$6010	$5918		$6011	$5244	
Median	$852	$889		$885	$786		$879	$959	
Mean pre-index inpatient costs	$827	$825	0.00	$927	$839	−0.01	$925	$846	−0.01
SD	$9156	$7774		$9997	$9983		$10,191	$8009	
Median	$0	$0		$0	$0		$0	$0	
Mean pre-index outpatient medical costs	$1734	$1696	−0.01	$1822	$1633	−0.03	$1812	$1624	−0.03
SD	$6347	$6621		$6523	$6409		$6463	$6493	
Median	$400	$397		$419	$371		$416	$361	
Mean TOTAL pre-index costs ^3^	$4958	$4951	0.01	$5221	$4795	−0.08	$5186	$4950	−0.02
SD	$13,353	$12,259		$14,116	$13,852		$14,214	$12,259	
Median	$2028	$2051		$2112	$1856		$2100	$2039	

^1^ SMD (absolute) ≥0.1 indicates significance; ^2^ Including: HIV/AIDS; metastatic cancer and acute leukemia; lung or upper digestive or other severe cancer; lymphatic, head, neck, brain, or major cancer; breast, prostate, colorectal, or other cancer; and disorders of immunity; ^3^ TOTAL = outpatient pharmacy + inpatient + outpatient medical; CCI = Charlson Comorbidity Index Score; IBD = Inflammatory bowel diseases (ulcerative colitis and Crohn’s disease); SMD = Standardized mean difference.

**Table 3 vaccines-08-00446-t003:** Economic outcomes—post propensity score matching (PSM) and generalized estimating equation model (GEE) adjustment.

Predicted Mean Annualized Cost	aTIV *N* = 234,313		TIV-HD *N* = 234,313		Incremental Mean
	Mean	95% CIs	Mean	95% CIs	
All-cause total	$9999	$9925–$10,069	$10,022	$9963–$10,088	$23.24
Influenza-related total	$28.21	$24.60–$32.39	$31.77	$27.73–$36.26	$3.56
Influenza-related hospitalizations	$27.59	$23.00–$32.85	$26.29	$22.41–$30.81	−$1.30
Influenza-related ER	$3.97	$3.52–$4.43	$4.49	$3.99–$4.97	$0.52
Influenza-related office visit + oseltamivir	$1.10	$1.04–$1.17	$1.36	$1.29–$1.45	$0.26

Influenza-related total = sum of (Influenza-related hospitalizations, Influenza-related ER, Influenza-related office visit + oseltamivir).

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
