# Peer review of "Evaluating the Relative Vaccine Effectiveness of Adjuvanted Trivalent Influenza Vaccine Compared to High-Dose Trivalent and Other Egg-Based Influenza Vaccines among Older Adults in the US during the 2017–2018 Influenza Season"

_vaccines, 2020, doi:10.3390/vaccines8030446_

Round 1
Reviewer 1 Report
The authors evaluate the rVE of aTIV compared to other egg-based influenza vaccines (TIV-HD, QIVe-SD, and TIVe-SD) against influenza-related hospitalizations/emergency room (ER) visits, influenza-related office visits, and cardio-respiratory hospitalizations/ER visits during the 2017-18 flu season in the elderly population in the United States; costs were compared between aTIV and TIV-HD subjects.
A linked subject population was utilized, using de-identified data from several IQVIA databases: Professional Fee Claims (Dx), Prescription Claims (Rx) and Hospital Charge Data Master (CDM). Subjects with ≥1 medical or pharmacy claim for aTIV, TIV-HD, QIVe-SD or TIVe-SD in Dx or Rx between August 1, 2017, and January 31, 2018 (the “selection window”) were initially identified.
The first claim determined the vaccine cohort and the date was termed the “index date”. Clinical outcomes of interest were assessed starting 14 days after the index date. The number and rates (events per 1,000 vaccinated-subject seasons) of the following events were assessed: influenza-related hospitalization/ER visits, influenza-related office visits, and cardio-respiratory hospitalization/ER visits. Cardio-respiratory hospitalization/ER visits related to the following events were assessed: pneumonia, asthma/COPD/bronchial, coronary artery (including myocardial infarction), congestive heart failure, cerebrovascular (including stroke), and other respiratory events.
Inverse probability of treatment weighting (IPTW) was used to adjust for confounders and treatment selection bias. Baseline characteristics and standardized mean differences post-IPTW were reported as a measure of balance. IPTW-weighted multivariate Poisson regression models were developed to estimate adjusted rate ratios (RR) along with corresponding 95% confidence intervals (CIs) for aTIV compared to the other vaccines in pairwise comparisons. Adjusted rVE was calculated as ([1-RR] * 100%) along with corresponding 95% CIs.
Finally, an economic analysis was conducted among aTIV and TIV-HD only. This comparison was considered relevant as these influenza vaccines are the only influenza vaccines with specific indications for subjects ≥65 years of age and that are primarily used in the elderly population in the US. Propensity score matching (PSM) was used to adjust for measured confounders. A greedy nearest neighbour matching technique without replacement at a ratio of 1:1 was performed, using calliper widths of 0.1 of the standard deviation of the logit of the propensity score. Baseline characteristics with SMD ≥0.10 were included in the match.
The authors found that aTIV was significantly more effective in preventing influenza-related office visits and other respiratory hospitalization/ER events compared to TIV-HD, QIVe-SD and TIVe-SD. aTIV was also associated with higher rVE against influenza-related hospitalization/ER visits compared to QIVe-SD and TIVe-SD, and was similar compared to TIV-HD aTIV was also associated with higher rVE against hospitalizations/ER visits related to pneumonia and asthma/COPD/bronchial events compared to TIVe-SD. All-cause and influenza-related total costs were comparable between aTIV and TIV-HD.
The authors have relevant stakes associated with those results as VD, MD and DS are employees of IQVIA which received funding for this study from Seqirus. SIP and MJP received financial support for time and effort from Seqirus for this study. Seqirus funded this study. GK and JM are employees and shareholders of Seqirus. GK and JM were involved in study design, interpretation of data, critical revision of the manuscript and in the decision to publish the results.
In my opinion, the approach, analysis, obtained results are well presented. That said, any improvement or the backbone of a good manuscript comes up with and in the Discussion.
I only have a few comments for the Discussion that I consider could contribute to improving the manuscript.
Overall the major limitation of real-world evidence in influenza studies at the current stage is that the estimates are performed own not laboratory-confirmed outcomes, this is a significant caveat that hopefully should be corrected in the future by having the possibility to include in the overall analysis a subanalysis restricted to all those with laboratory test performed and by the result.
This evident fact is to be emphasized and recognized and not glossed over in a couple of lines (see 483-484). This is a significant limitation.
One consequence is that the actual season virological description is tiptoed over. So my first comment is a recommendation to improve that aspect of the Discussion.
Page 15, lines 393 to 394. I suggest to describe more in length the virological characteristics of this particular season in the US, with the first wave of A(H3N2) and the second wave of B/Yamagata-lineage, and discuss the egg-derived effect and mismatch B effect of circulating viruses n the outcomes of interest, how could they have had an impact on the included vaccine effectiveness, and the rationale for the observed favourable differences in vaccine effectiveness in this particular situation.
Lines 411 and 415. The authors write:
“While we followed similar definitions and methods, our sample was not restricted to Medicare FFS patients, and post-IPTW, approximately one-third of our cohorts had a third-party payer. While we can identify patients with Medicare or Commercial payer type in the utilized data sources, we are unable to distinguish those with Medicare Advantage. “
Please, clarify why and how this broad “payers” landscape explains or could have and impact on the contrary results between the current results reported by the authors and the results reported by Izurieta et al. (2019). Additionally, the authors mention the same “note single-payer” argument in lines 493-494. So that seems an important point, please clarify and support by this is so pivotal.
This is important due that the differences with the Van Aalst et all (2020) paper are well discussed.
Finally, this is a multiple-comparison study, in which not fewer than 30 rVE estimates are reported. This is not mentioned and neither discussed. I recommend the authors, that say on line 491 that they “provide a conservative estimate”, to argue why they have not performed a “multiple-comparison” correction of their estimates.
Reviewer 2 Report
The authors performed the retrospective analysis to compare the vaccine efficacy in elderly people of the adjuvanted trivalent influenza vaccine with that of the high-dose trivalent, standard-dose trivalent, and standard-dose quadrivalent influenza vaccines. They found that the adjuvanted trivalent influenza vaccine showed a higher vaccine efficacy than other vaccines.
Overall the experimental design and execution are reasonable. However, several minor revisions are required to improve this manuscript.
Specific comments:
- Figure 1. The horizontal arrows should be erased.
- Figure 5. The bar and its group name are misaligned.
Reviewer 3 Report
In this manuscript, Pelton et al. evaluate the relative vaccine effectiveness of different influenza vaccines using a retrospective cohort study. They find that adjuvanted trivalent influenza vaccine (aTIV) was more effective in reducing influenza-related office visits and other respiratory-related hospitalization visits compared to other vaccines in subjects aged 65 years and older. My comments are minor.
Specific Comments:
“Real-world” should be removed from the document (title, lines 88, 380, 396). The rest of the title/description of retrospective cohort study implies that this is not an in vitro or animal study.
The abstract needs to end with an overall conclusion/importance sentence(s) (for the entire study).
The letters/numbers are blurry in many of the figures. This should be adjusted.
Lines 398-410 and 426-438 are missing many citations, as the authors are citing specific numbers/findings from the Izurieta or Van Aalst studies.
